# Register and `[CLS]` tokens induce a decoupling of local and global features in large ViTs

**Alexander Lappe**[1,2]    **Martin A. Giese**[1]
[1]Hertie Institute, University Clinics Tübingen    [2]IMPRS-IS
`alexander.lappe@uni-tuebingen.de`

## Abstract

Recent work has shown that the attention maps of the widely popular DINOv2 model exhibit artifacts, which hurt both model interpretability and performance on dense image tasks. These artifacts emerge due to the model repurposing patch tokens with redundant local information for the storage of global image information. To address this problem, additional register tokens have been incorporated in which the model can store such information instead. We carefully examine the influence of these register tokens on the relationship between global and local image features, showing that while register tokens yield cleaner attention maps, these maps do not accurately reflect the integration of local image information in large models. Instead, global information is dominated by information extracted from register tokens, leading to a disconnect between local and global features. Inspired by these findings, we show that the `[CLS]` token itself leads to a very similar phenomenon in models without explicit register tokens. Our work shows that care must be taken when interpreting attention maps of large ViTs. Further, by clearly attributing the faulty behavior to register and `[CLS]` tokens, we show a path towards more interpretable vision models.

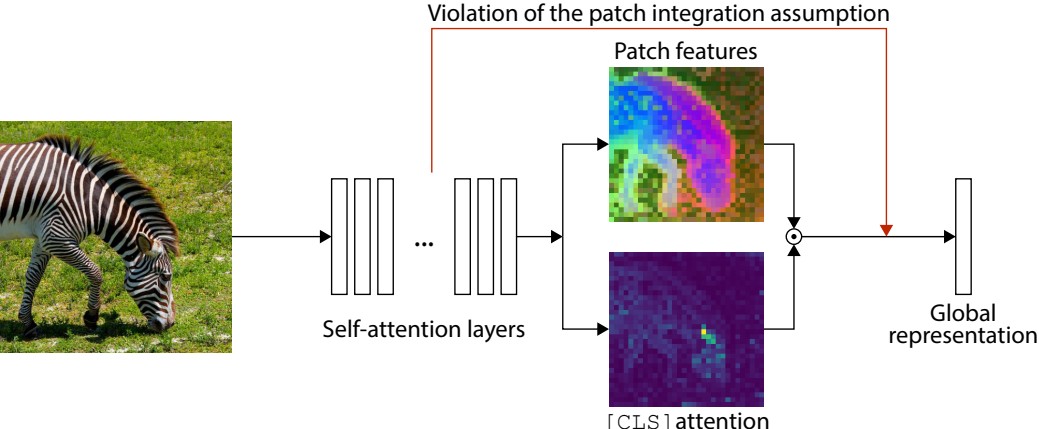

Figure 1: The global image representation computed by a Vision Transformer is usually understood as a weighted average of the local patch features, where weights are given by the `[CLS]` token attention scores. This notion, which we refer to as the *patch integration assumption*, underlies attribution methods like plotting the attention maps to identify patches that strongly contribute to the global output. In this work, we show that both register tokens and the `[CLS]` token lead to violations of this assumption in large ViT models.

39th Conference on Neural Information Processing Systems (NeurIPS 2025).

# 1 Introduction

In recent years, the application of the Transformer architecture [1] to computer vision [2] has given rise to powerful, highly general feature extractors which can be used for numerous downstream vision tasks. These models are pretrained on large datasets in either supervised [3, 4, 5, 6] or self-supervised [7, 8, 9, 10] fashion, and then fine-tuned or used directly as off-the-shelf feature extractors. A useful property of the Vision Transformer architecture is that it simultaneously extracts global and dense local features, making it suitable for both classification, as well as more fine-grained tasks such as object detection [11], segmentation [12] or depth estimation [10, 13]. Further, the fact that the global image representation is computed as a convex combination of the dense features enables direct comparisons between patch and global representations [14, 15, 16]. The weights used for this convex combination are the attention outputs of the global [CLS] token to the patch tokens. Caron et al. [7] popularized the investigation of the attention scores as a way of attributing model outputs to image regions, as one can seemingly clearly identify the patches from which the global information is extracted. After the publication of the now widely used DINOv2 model [10], Darcet et al. [17] showed that the model's attention maps exhibit artifacts with very large weights. Upon further investigation, they noted that these artifacts also appear in other models trained with different training strategies, and that the patch tokens corresponding to the artifacts contain global image information rather than information about the image patch as originally assumed. Since this phenomenon made the attention maps less interpretable and degraded the model's performance on dense tasks, the authors proposed the introduction of additional register tokens in which global information can be stored, to prevent it from being encoded in the patch tokens. The inclusion of such register tokens achieved the desired effect of removing artifacts from the attention maps. Further, while there has been some debate on how informative attention maps are in general [18, 19], the resulting attention maps have also been argued to be more interpretable [17, 20]. Subsequently, further work proposed alternative ways of including register tokens in the model without repeating the expensive pretraining stage [20, 21].

However, the encoding of global information in the patch tokens points towards a deeper underlying problem. Even if this information is stored in register tokens instead, it is unclear whether the denoised attention maps over the patch tokens actually reflect the global model output faithfully. Since evidently some global information is integrated before the last layer and can flow from the register tokens to the [CLS] token, it must be studied to which degree the patch features still contribute to the final output. In this work, we show that in large models, the global image representation relies primarily on information extracted from the register tokens, which needs to be considered when studying the denoised attention maps. This effect does not occur in the smaller model variants, which display a tight correspondence between local and global features. Inspired by this finding, we examine whether a disregard of the local patch features can also occur in overparameterized models without explicit register tokens. We observe that the [CLS] token itself may be interpreted as a register and show that the implicit backward-attention mechanism introduced by the residual connection leads to the same effect.

**Contributions.** Our results show that the intuitive correspondence between global and local image representations holds in smaller variants of the ViT trained with DINOv2, but breaks down for larger models. Importantly, the last-layer attention maps do not faithfully represent the mechanism through which global image representations are formed in large DINOv2 models with register tokens or residual connections in the [CLS] token. These effects need to be taken into account when studying attention maps to attribute model outputs to image regions, or ground global feature representations in local patch features. Our findings suggest that future iterations of generalist vision models should be built without register tokens and residual connections in the [CLS] token to ensure model interpretability.

# 2 Preliminaries

**Vision Transformers.** Given an image $\mathbf{x} \in \mathbb{R}^{h \times w \times 3}$, a vanilla Vision Transformer divides it into small patches and feeds these patches through a network of self-attention layers. This process yields a feature map $\mathbf{z} \in \mathbb{R}^{p_1 \times p_2 \times d}$, where $p_1$ and $p_2$ denote the spatial dimensions and $d$ the model's feature dimension. To learn a global representation of the image, an additional [CLS] token is introduced, which is treated exactly like the image patch tokens. The output of the [CLS] token at the last layer corresponds to the global image embedding. Mathematically, the self-attention mechanism for a

single attention head works as follows: The [CLS] token outputs a query vector $\mathbf{q} \in \mathbb{R}^m$ which is compared to the keys output by each token denoted by $\mathbf{k}_1, \ldots, \mathbf{k}_{p_1 p_2 + 1} \in \mathbb{R}^m$ to yield the attention vector

$$\mathbf{a} := \mathrm{softmax}(\langle \mathbf{q}, \mathbf{k}_1 \rangle, \ldots, \langle \mathbf{q}, \mathbf{k}_{p_1 p_2 + 1} \rangle). \tag{1}$$

The output of the attention mechanism for the [CLS] token is then given by the convex combination

$$\mathbf{o}_{cls} := \sum_i a_i \mathbf{v}_i, \tag{2}$$

where $\mathbf{v}_i$ denotes the value vector of the $i$-th token. The final processing after the attention mechanism differs slightly between models. In the DINOv2 model we consider here, the attention is followed by a residual connection, layer normalization and a shallow multi-layer perceptron to yield the final output. As the residual connection is applied after self-attention and does not affect the features extracted via the attention map, we exclude it from the computations when comparing contributions of register tokens and patch tokens in Section 3. We treat the influence of the residual connection separately in Section 4.

**The patch integration assumption.** Vision Transformers are therefore usually understood as patch feature extractors, where the [CLS] token learns to select the final features from the most relevant patches for global image understanding. We refer to this notion as the *patch integration assumption*. Based on the fact that the [CLS] token undergoes the same transformations as the patch tokens and can be written as a weighted average thereof, the patch integration assumption has been used to attribute global model behavior to specific image regions [14, 15, 16]. In particular, the attention vector $\mathbf{a}$ of the [CLS] token is a popular object of study to determine the patches that the model relies on for its global image representation [2, 7, 22, 23, 24]. A body of work on interpretability of ViTs has suggested alternative methods for attributing model behavior, going beyond studying the [CLS] attention scores [25, 26, 27, 28]. However, only Darcet et al. [17] challenged the *patch integration assumption* itself by observing the emergence of high-norm tokens in large models.

**Transformers with register tokens.** As mentioned previously, Darcet et al. [17] found that very large models trained with DINOv2 store global image information in patches that otherwise contain redundant information, degrading the quality of the dense patch features. As a remedy, they introduced additional register tokens that are appended to the patch tokens just as the [CLS] token. These are supposed to be used for storing global information to avoid the use of the patch tokens themselves for this purpose. In this case, the output of the [CLS] token becomes

$$\mathbf{o}_{[CLS]} = \underbrace{\sum_{i \in \{\text{patches}\}} a_i \mathbf{v}_i}_{\text{patch contribution}} + \underbrace{\sum_{j \in \{\text{registers}\}} a_j \mathbf{v}_j}_{\text{register contribution}}. \tag{3}$$

**Decoupling of patch and global features.** Darcet et al. [17] show that the resulting attention maps of DINOv2 models including registers are less noisy, and the resulting patch features show better performance on dense prediction tasks. However, it is possible that the inclusion of registers decouples the global image embedding from the local patches, as the [CLS] token is no longer a convex combination of only the local features. In case the [CLS] token attends primarily to the register tokens, the attention maps on the patches, while shown to be less noisy, might be uninformative due to dominance of non-local features. Further, if global information is obtained from the registers instead of integrating information from the patches, feature spaces between local and global representations might not be aligned, hurting the ability to precisely specify the local grounding of global information [14, 15, 29].

## 3 The influence of register tokens on the patch integration assumption

### 3.1 Larger models attend more to register tokens and less to patch tokens

High-norm patch tokens have so far only been observed in the larger variants of the ViT architecture. This raises the question of whether only larger models rely heavily on additional register tokens. To answer this question, we test how much attention the last-layer [CLS] token places on the register

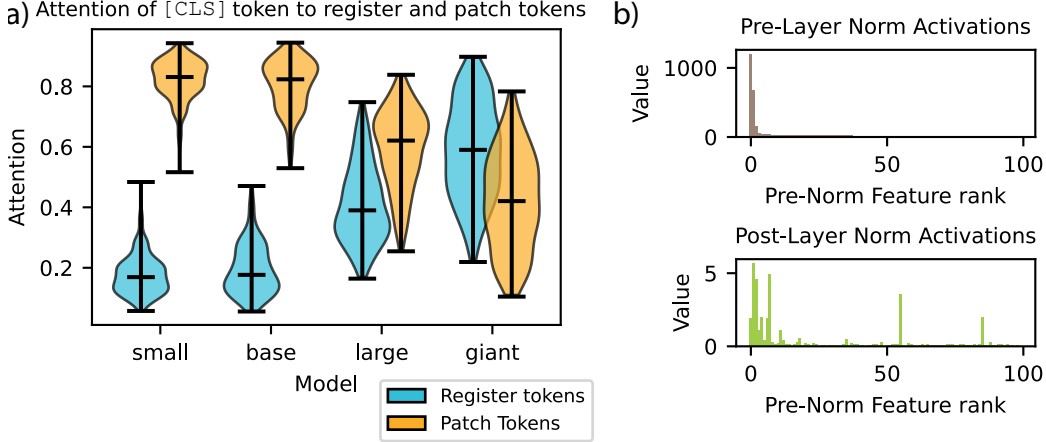

Figure 2: **a)** The amount of attention placed by the `[CLS]` token of the last layer onto the patch and register tokens, respectively. Smaller models attend primarily to the patch tokens, whereas bigger models attend more strongly to the register tokens. **b)** Mean activations of the highest-norm register token in the last layer of the 'giant' model. The 100 dimensions with highest activations before the layer norm are shown. Register tokens show large activations in a small subspace, making them seemingly image-independent as measured by pairwise cosine similarity (top panel). However, the layer norm downscales these dimensions *before* the self-attention mechanism (bottom panel).

and patch tokens, respectively. We study the DINOv2 models with register tokens as published by the authors on huggingface.com and probe them using the validation images of the MS COCO dataset [30]. We study the 'small', 'base', 'large' and 'giant' models, which differ in number of self-attention layers, hidden dimension and number of attention heads. They have $21M$, $86M$, $300M$, and $1,100M$ total parameters, respectively. We extract the post-softmax attention vectors of the `[CLS]` token as presented in Eq. (3), average them across attention heads and then sum the resulting scalars for the register tokens and the patch tokens respectively, to examine how attention is partitioned between the two types of tokens. For each image, the sum of register attention and patch attention is one. Results are shown in Fig. 2.

We observe clearly that smaller models attend primarily to patch tokens, and the 'large' and 'giant' variants place more attention on the register tokens. Strikingly, the variance of preference for either register or patch tokens in the largest model is considerable, with some images attending mainly to patch tokens, and some almost ignoring them entirely.

### 3.2 Information in register tokens is only seemingly image independent

**Image dependency of high-norm tokens.** The role of information encoded by high-norm tokens in the original DINOv2 model is not entirely clear. Darcet et al. [17] trained linear probes on the high-norm tokens, concluding that they contain more global information and less positional information than the regular patch tokens. On the other hand, Wang et al. [31] claimed that high-norm tokens are image independent and can be predicted by the first singular vector of a linear approximation of the attention layer itself. They support these findings by noting that the pairwise cosine similarity between high-norm tokens of different images is extremely high. This conclusion raises the question of why the `[CLS]` token of the original DINOv2 model would be subject to image-independendent information. By studying representations *within* the self-attention layer, we first reveal that these previous findings also apply to the register tokens, corroborating the assumption that they indeed take over the role of high-norm patch tokens. Further, we show that the previous results are not at all at odds, and that high-norm register tokens do carry image-dependent information.

**Influence of layer norm.** To understand in how far register tokens encode image-dependent information, we study their hidden states at the penultimate layer of the 'giant' model. The penultimate layer is the most relevant, since the final `[CLS]` output is formed by attending to these representations. First, we note that the hidden states of the register tokens do have abnormally high norm, and that the

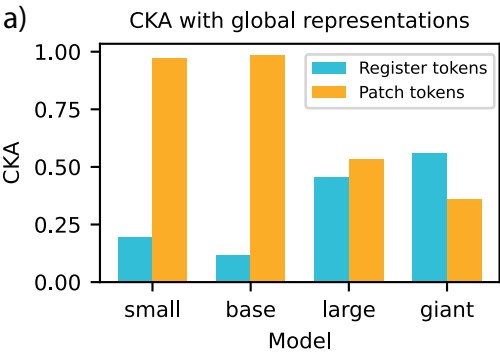 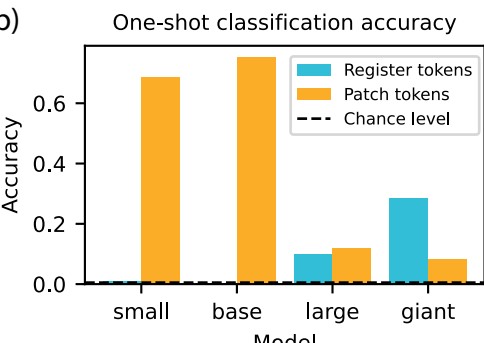

Figure 3: **a)** Centered kernel alignment between the global `[CLS]` token output, and `[CLS]` token output computed while only attending to either register tokens or patch tokens. Patch tokens yield a faithful representation of the global output for smaller models, but the connection between local and global features breaks down with increasing model size. **b)** One-shot classification accuracy on the 1000 Imagenet classes. The classifier is trained on the global `[CLS]` output and then tested on output based on patch and register tokens, respectively. Attending only to the patch tokens yields poor performance in the larger models, corroborating the finding that global representations are not formed by attending to the patch features.

same token has the highest norm among all tokens for any input image by an order of magnitude. Second, we observe that the finding from Wang et al. [31], that activations are very similar across images, hold for the register tokens as well. The average pairwise cosine similarity between images in the highest-norm token is 0.9989, seemingly confirming that these tokens are image independent. However, studying the feature activations more closely shows that this need not be the case. Fig. 2 shows the top-100 feature dimensions of the highest-norm token (features with highest activation), averaged across all images. The plot clearly shows that the vector is dominated by extreme outliers which in turn dominate the cosine similarity. This leads us to two important points: First, note that the hidden state itself does not appear in the self-attention mechanism via a dot-product comparison. Instead, keys and values, which *are* compared via the dot product, are computed as linear maps of the hidden state, allowing the model to ignore these dimensions and focus on potentially image dependent dimensions instead. This mirrors exactly the methodological differences between [17] and [31], demonstrating why the authors came to opposite conclusions. Further, before entering the self-attention mechanism, the ViT applies a layer norm to all tokens, thereby reweighting the features and downscaling high-norm tokens. Fig. 2 shows the same data *after* applying the layer norm, showing that outlier dimensions are scaled down before computing self-attention.

### 3.3 Registers tokens dominate global image representations in large models

So far, we have shown that larger models attend strongly to register tokens, and we have discussed how register tokens can encode image-specific information. It remains to study how strongly the register tokens actually influence the output of the self-attention mechanism. After all, the information extracted by register tokens might be redundant w.r.t. to the patch tokens, or their value vectors could have low norm, leading to weak contribution despite large attention.

To examine the influence of patch tokens and register tokens on the `[CLS]` output, we constrain the final self-attention layer to attend to only one of the two token types. Specifically, we set all attentions to one of the two token types to zero, yielding a `[CLS]` output that is based only on the other. We also compute the unaltered model output based on all tokens and investigate the similarity of the resulting global representations using linear centered kernel alignment (CKA) [32]. Given two (mean-centered) matrices of neural network activations $X \in \mathbb{R}^{n \times d_1}$, $Y \in \mathbb{R}^{n \times d_2}$ computed on $n$ test samples, linear CKA is defined as

$$CKA(X,Y) = \frac{\text{tr}(XX^\intercal YY^\intercal)}{\sqrt{\text{tr}(XX^\intercal XX^\intercal)\text{tr}(YY^\intercal YY^\intercal)}} \tag{4}$$

and can be interpreted as measuring the alignment of pairwise similarities between the two matrices. Fig. 3 a) shows the CKA between the unaltered output, and the patch-based and register-based output,

respectively. CKA was again computed using the samples from the MS COCO test set. We observe that for the two smaller models, the patch-based representation shows close-to-perfect alignment with the `[CLS]` output. This indicates that the notion of the global output being an aggregate of last-layer local patch features is accurate. However, with increasing model size, the global output and the patch-based output become highly disconnected, and outputs based on register tokens represent the global output more faithfully in the giant model. The same analysis for the Imagenet test set [33] is presented in the appendix, mirroring the qualitative results. Fig. 3 b) displays one-shot top-5 classification accuracy when training the classifier on the global `[CLS]` output and evaluating on either the patch-based or register-based output. Results were generated by randomly sampling one training image and one test image per class from the 1000 classes in the Imagenet test set. In accordance with the results from panel a), we observe that classifiers relying only on patch features perform poorly for the two larger models. This demonstrates that class-relevant global information is *not* extracted from the patch features.

### 3.4 Attention maps with registers are clean but do not reflect global image representations

The original purpose of including register tokens in the DINOv2 model was to remove high-norm tokens from the image patches to obtain better dense representations and cleaner attention maps. However, Fig. 3 casts doubt on how informative the cleaned-up attention maps are if the patch features extracted via the attention maps do not accurately represent the global model outputs. We show an example in Fig. 4. As expected, we observe that the attention map output by the giant model with registers is visually cleaner than the one output by the vanilla model. Next, we examine how faithfully the features extracted according to this attention map represent the total layer output by computing the cosine similarity between the extracted patch features, and the total layer output including register tokens. The cosine similarity is -0.0092, meaning that the features extracted according to the attention map are completely orthogonal to the features extracted by the `[CLS]` token when including the register tokens. While not all images elicit disconnects of this magitude, the example demonstrates the violation of the *patch integration assumption* in large models with registers, showing that its attention maps should not be relied on.

At the global level, let us consider the connection between the attention maps over patches and the CKA results shown in Fig. 3. The purpose of the `[CLS]` token attention is to extract global image features that facilitate downstream tasks. Relevant for task performance is how images are embedded *relative* to each other, which, as is done in CKA, can be measured by the Gram matrix $\mathbf{X}\mathbf{X}^{\intercal}$, where $\mathbf{X} \in \mathbb{R}^{n \times d}$ contains the feature representations of a set of test samples. This Gram matrix encapsulates the similarity structure of learned image representations. Writing $\mathbf{X} = \mathbf{X}_p + \mathbf{X}_r$ as the sum of patch contributions and register contributions Eq. (3), we can decompose the Gram matrix as

$$\mathbf{X}\mathbf{X}^{\intercal} = \mathbf{X}_p\mathbf{X}_p^{\intercal} + \mathbf{X}_r\mathbf{X}_r^{\intercal} + \mathbf{X}_p\mathbf{X}_r^{\intercal} + \mathbf{X}_r\mathbf{X}_p^{\intercal}. \tag{5}$$

The attention maps only give insight into which patches are relevant for computing the patch-based representation and thus embedding an image into the representational geometry determined by $\mathbf{X}_p\mathbf{X}_p^{\intercal}$.

However, Fig. 3 shows that the geometries of $\mathbf{X}_p\mathbf{X}_p^{\intercal}$ and $\mathbf{X}\mathbf{X}^{\intercal}$ are misaligned, indicating that the attention map does not yield sufficient insight into how an image is embedded in the models's global representational geometry.

### 3.5 Connections to overparameterization and neural collapse

The analysis of Darcet et al. [17] showed that high-norm patch tokens only appear in the larger variants of the ViT architecture. Our results on register tokens are similar, showing that both `[CLS]` token attention to registers, as well as their influence on the global image representations increase substantially with model size. These findings relate to literature on how representations are formed in overparameterized models. Work on neural collapse [34] demonstrates that models with large capacity tend to learn simple, very low-dimensional representations at the last layer and that such representations yield better performance [35, 36, 37] and generalization abilities [34]. Further, it has been shown for overparameterized classification models, that low-intrinsic dimensionality as well as linear separability of classes emerge already before the final layer [38, 39]. Integrating our findings with this body of work, we hypothesize that larger ViT variants are overparameterized to the point at which DINOv2 training yields simplistic representations at the last layer, relying on a small number of tokens in which global information has already been integrated. The introduction of

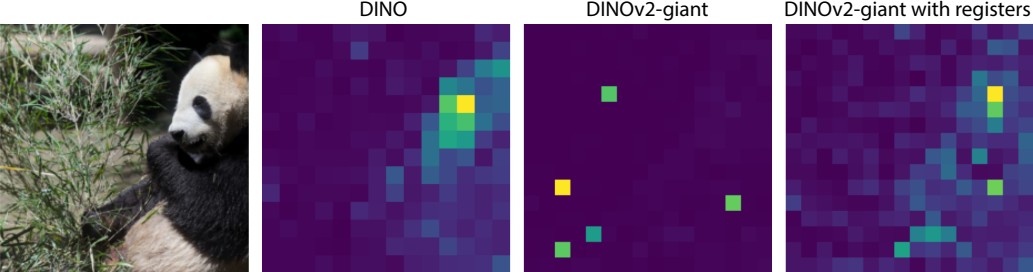

Figure 4: Attention maps of the final [CLS] token. A convex combination of the corresponding patch features yields the patch-based contribution to the global image representation. As noted by Darcet et al. [17], the attention map of the DINOv2 model exhibits large artifacts. These are removed by including register tokens in the model, seemingly leading to a more interpretable attention map. However, when computing the [CLS] output based on the convex combination of patch features in the model with registers, its cosine similarity to the total output of the final layer is -0.0092. In other words, attending to the patch tokens yields a representation completely orthogonal to the one including the register tokens, showing that the attention map fails to attribute global information to image patches.

register tokens merely shifts this mechanism outside of the patch features, preserving simple last-layer representations in the register tokens and thus still allowing the model to violate the *patch integration assumption*.

# 4 The influence of the [CLS] token on the patch integration assumption

## 4.1 Very large models attend primarily to features from the skip connection

So far, we have discussed how register tokens lead to disconnects between local and global image representations and thereby the violation of the patch integration assumption. We have hypothesized that this behavior is due to overparameterization, allowing the model to solve its image-level training objective before the final layer, and storing the information in the register tokens. This line of reasoning motivated an inquiry into the mechanisms through which similar behavior could arise in overparameterized models without explicit register tokens. We have already discussed how the original DINOv2 model adapts patches containing little local information to instead store global information. In this section, we explore another mechanism that allows the model to ignore patch information at the last layer, namely residual connections within the [CLS] token.

**Attending to the previous layer.** Previous work has shown that overparameterized networks may form more or less final representations of the input relatively early in the model hierarchy [38, 39]. Since the [CLS] token in the ViT is appended to the patch tokens before the first layer, the standard ViT already contains one register token, which can be used to store emerging global information. Importantly, recent implementations of ViTs include skip connections within the attention block, resulting in the [CLS] token output

$$\mathbf{o}_{[CLS]} = \sum_i a_i \mathbf{v}_i + \mathbf{o}_{[CLS]}^{prev} = \underbrace{\sum_{i \in \{patches\}} a_i \mathbf{v}_i}_{patch\ contribution} + \underbrace{a_{[CLS]} \mathbf{v}_{[CLS]} + \underbrace{\mathbf{o}_{[CLS]}^{prev}}_{skip\ contribution}}_{non\text{-}patch\ contribution}, \qquad (6)$$

where $\mathbf{o}_{[CLS]}^{prev}$ denotes the output of the [CLS] token at the previous layer, and $a_{[CLS]}$ and $\mathbf{v}_{[CLS]}$ denote the attention value to the [CLS] token and its value vector, respectively. This allows the output to attend to the [CLS] token via two different mechanisms: The first is part of the standard self-attention as given in equations (1) and (2). The second is given implicitly through the relative scales of the contributions of the attention mechanism and the skip connection to the output $\mathbf{o}_{[CLS]}$ as given in Eq. (6). These relatives scales depend on the model parameters; therefore the model learns during training how strongly to attend to the skip connection.

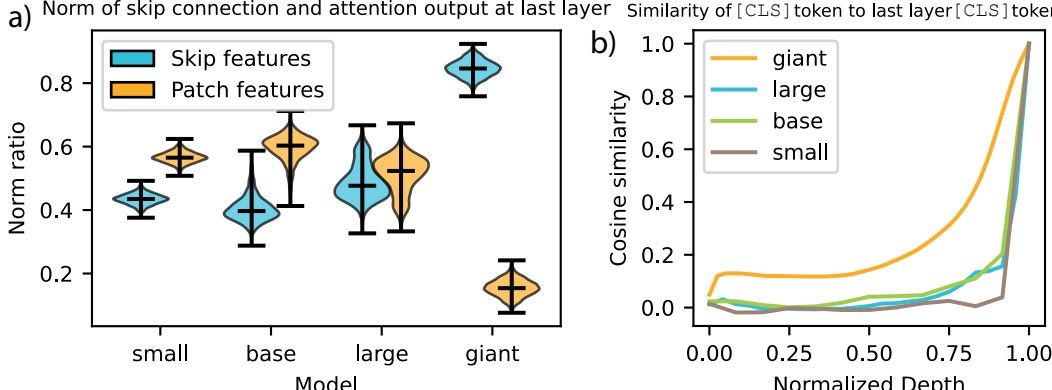

Figure 5: a) After the self-attention mechanism, a skip connection sums the attention output and the hidden states from the previous layer, providing an alternative way for the `[CLS]` token to attend to itself. Since the attention weights are given only implicitly, we plot the $L_2$-norm of the contributions of the skip connections and the patch features to the `[CLS]` token as a proxy. We observe that in the 'giant' model, the global output is primarily determined by previously computed features, rather than the patch features. b) We show the cosine similarity of the `[CLS]` token at all model layers to the `[CLS]` token at the last layer. The three smaller models exhibit a large jump at the very last layer, indicating that the `[CLS]` token at the last layer is strongly influenced by the patch tokens. Conversely, the `[CLS]` token of the 'giant' model converges to the final output more smoothly, explaining its low reliance on the patch features at the last layer.

**Measuring attention to patch features and skip features.** To examine how strongly models without register tokens attend to the skip connection, we study the original implementation of DINOv2 without registers, as well as DeiT III [6] for a supervised model, CLIP [3] for language supervision, and iBOT [9]. For readability we present findings for DINOv2 in the main text and show the generality of the findings using the other models in the Appendix. Since attention weights for the skip connection are not computed explicitly, we compute as a proxy the $L_2$-norms of the patch contribution and the skip contribution from Eq. (6). The results of this analysis on the MS COCO validation set are shown in Fig. 5. We observe that the three smaller models attend to both patch and skip features, but the 'giant' model output is strongly dominated by the skip features. Panel b) sheds further light on this phenomenon: The similarity of the `[CLS]` token of the last layer and all previous layers is low for the smaller models, indicating that the final `[CLS]` output is determined by integrating the final patch features. This is in line with the patch integration assumption. However, the structure of the output of the 'giant' model emerges earlier, and the penultimate output is already very similar to the

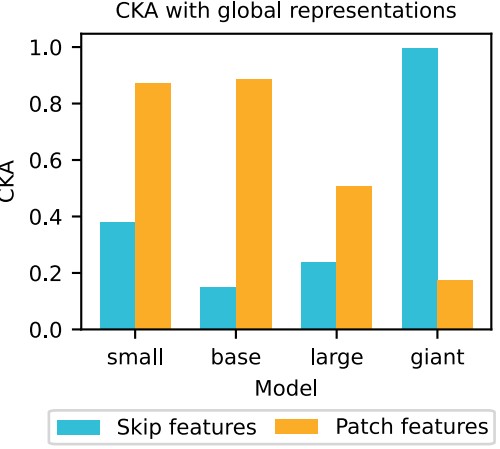

Figure 6: Centered kernel alignment between the standard `[CLS]` token output, and `[CLS]` token output computed using either only patch token features or the skip connection features from DINOv2. We observe that the findings from Fig. 5, that the giant model attends primarily to the skip connection, does indeed lead to disconnected representations between global features and patch features. This shows that the skip connection for the `[CLS]` token can by itself lead to a violation of the patch integration assumption.

final one. Therefore, the last layer receives the quasi-final output through the skip connection and only needs to attend weakly to patch features to compute small corrections. We show that other ViT models trained with different recipes mimic this behavior in the Appendix (Fig. 7).

Earlier, we found for models with register tokens that large attention to registers leads to a disconnect between local and global image features. Does the same hold for large attention to skip features? To answer this question, we compute the centered kernel alignment between the global model output, and model output based only on patch or skip features, respectively. The results are shown in Fig. 6 (results for other models are presented in the Appendix, see Fig. 8). Again, we find that for the larger models, the features extracted from the patches do not yield a faithful representation of the global model output. We conclude that the skip connection of the [CLS] token indeed disconnects the global output from the patch tokens in the larger models.

## 5    Discussion

A tight correspondence between local and global image features computed by Vision Transformers is desirable for both interpretability and tasks combining local and global image information like object detection. For smaller DINOv2 models, including the widely used 'base' model, we found no evidence for disconnects between local and global features which is in line with previous findings regarding high-norm tokens [17]. On the other hand, we have observed that the global output of 'giant' model variants does not correspond to the final patch features extracted via the self-attention mechanism in models with registers, making their attention maps unreliable. Therefore, we hypothesize that these disconnects arise as a consequence of overparameterization, which gives models the flexibility to integrate global information already in intermediate layers. Combining results from Darcet et al. [17] with the work at hand, we identify three mechanisms through which the *patch integration assumption* may break down in overparameterized models. First, patch tokens can be repurposed to store prematurely emerging global information. Second, an approach of alleviating the first issue based on introducing register tokens seems to only move the problem outside of the patch tokens, still allowing the model to disregard patch features. Third, the skip connection in the self-attention layer enables the [CLS] token to extract image representations gradually, resulting in the fact that the last attention layer does not extract features that accurately represent the final output.

**Future design choices.**    We have shown that while register tokens remove artifacts from the patch representations, they do not fully solve the problem of degenerate attention maps as they dominate the final attention-layer's output. This effect is particularly strong for the [CLS] token itself, which can be interpreted as a register token with an especially strong bias to attend to itself via the residual connection. We therefore argue that models that satisfy the *patch integration assumption* should be built without register tokens and hidden-layer [CLS] tokens. Another approach might be explicit regularization, although experiments on DeiT III show that drop-path regularization on it's own does not prevent the violation of the *patch integration assumption*. In any case, it remains to ensure that global information is not stored in repurposed patch tokens. Here, approaches that impose direct regularization on the patch tokens appear to be the most promising. Along those lines, Wang et al. [31] proposed enforcing patch tokens to have similar representations as their neighbours, effectively removing high-norm patch tokens without the introduction of an additional mechanism for storing global information. The authors test their approach by finetuning on a smaller dataset and show that the resulting model performs well on dense tasks.

**Limitations and future work.**    Our analysis focuses on diverging global encoding geometries between register tokens and patch tokens. Since the variance over samples with respect to the registers is high (see Fig. 2), future work could study the properties that lead an image to be processed primarily by patch/register tokens. As we have seen that centered kernel alignment between register tokens and patch tokens is low, a systematic analysis of the respective directions of variance in the representations could shed further light on the effects we have discussed. Since both attention to the skip connection and attention to explicit registers emerge in larger models, a more focused inquiry into the interdependence of these effects would be interesting. Finally, it remains to study the impact of violations of the *patch integration assumption* on performance on global image tasks. As mentioned in Section 3.5, previous work has shown a relationship between simplistic last-layer embeddings and model performance. A comparison of complexity-matched models that are forced to satisfy the *patch integration assumption* and models with registers is necessary for determining whether there is a tradeoff between image-level task performance and correspondence of local and global features.

**Broader impacts.** We do not foresee malicious use of our work, or tangible potential negative impact as a whole. On the contrary, we hope that our contribution will aid in better understanding of foundation vision models.

# 6 Acknowledgements

AL and MG are supported by ERC-SyG 856495. MG is supported by HFSP RGP0036/2016, BMBF FKZ 01GQ1704. The authors thank the International Max Planck Research School for Intelligent Systems (IMPRS-IS) for supporting Alexander Lappe. Finally, we are grateful to Vojta Smekal, Marta Poyo Solanas, Prerana Kumar, and Lucas Martini for valuable discussions and support.

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

# A   Supplementary Results

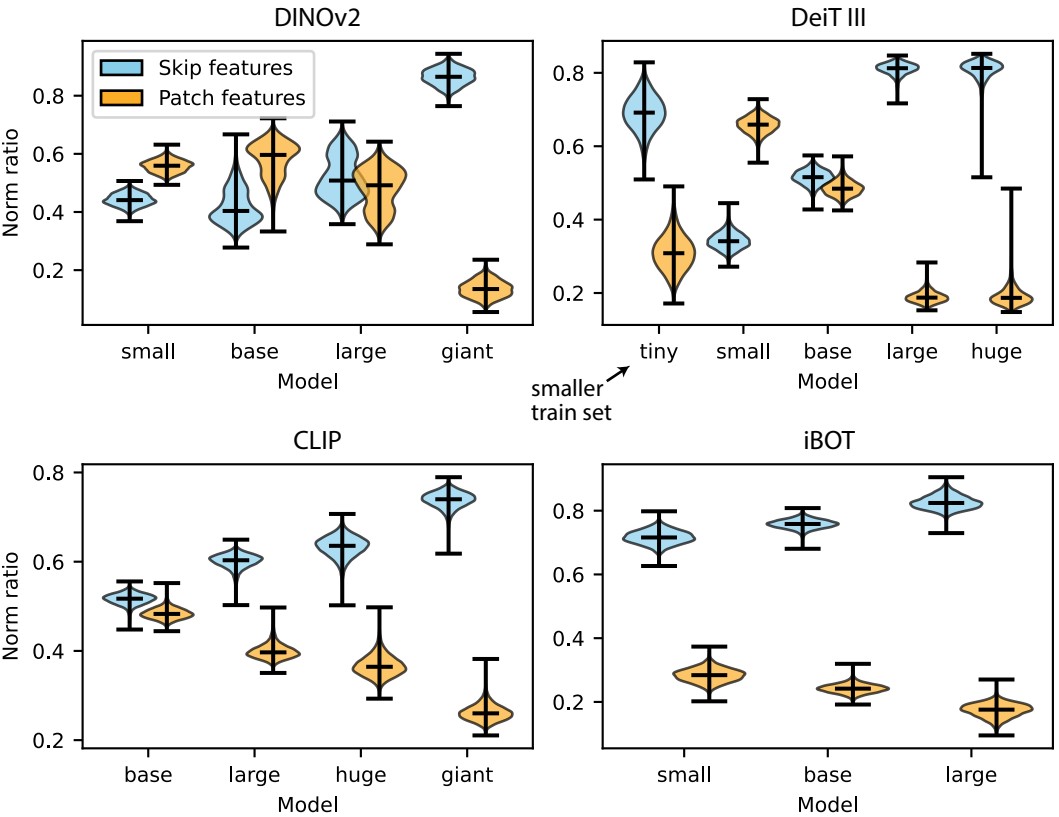

Figure 7: Attention to the `[CLS]` token as discussed in Section 4, for Vision Transformers trained with different recipes. Note that attention to the skip connection consistently grows with model size. For DeiT III, we also include the tiny model, which was trained on a smaller data set. Interestingly, the tiny model mimics the behavior of the huge one trained on more data, demonstrating the importance of the the relationship between model capacity and data set size as discussed in the main text.

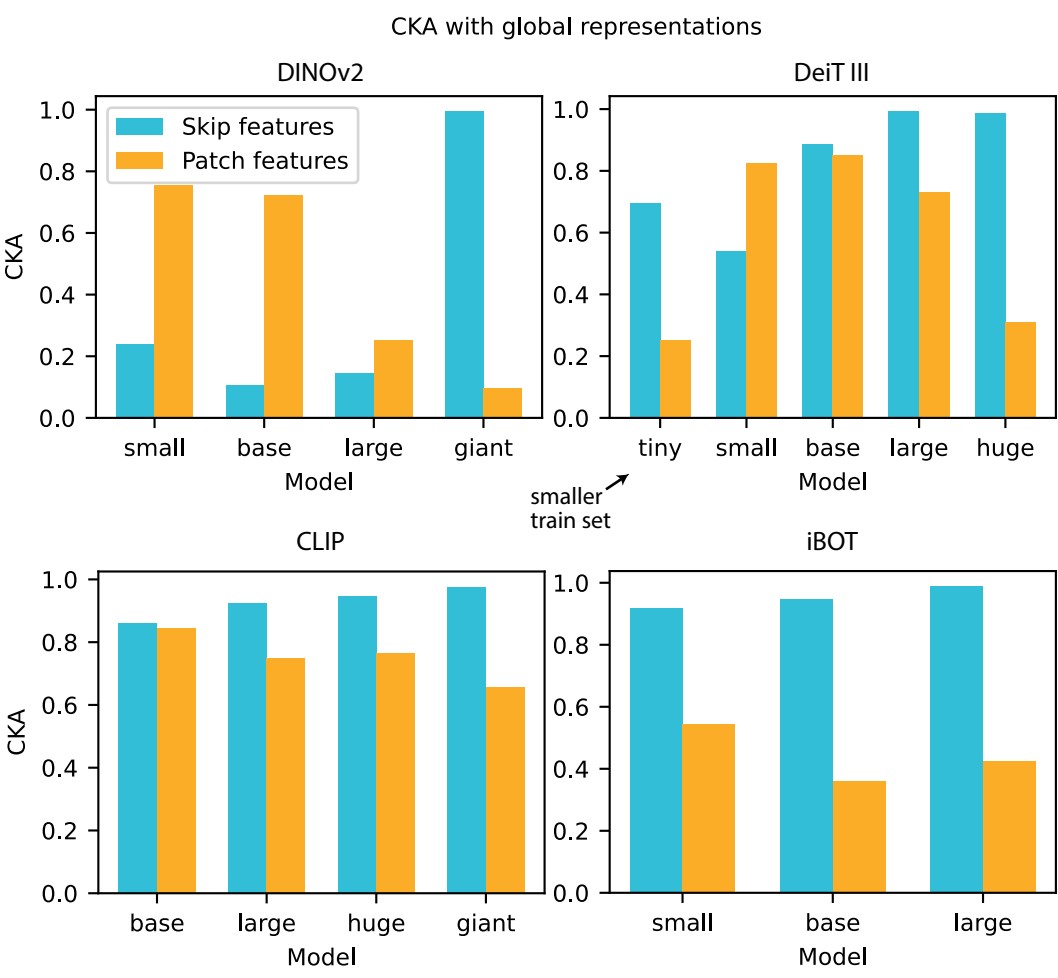

Figure 8: CKA results for the skip connection as discussed in Section 4 for different pretraining recipes. A disconnect between the local and global image features as discussed in the main text occurs across all pretraining recipes with growing model size.

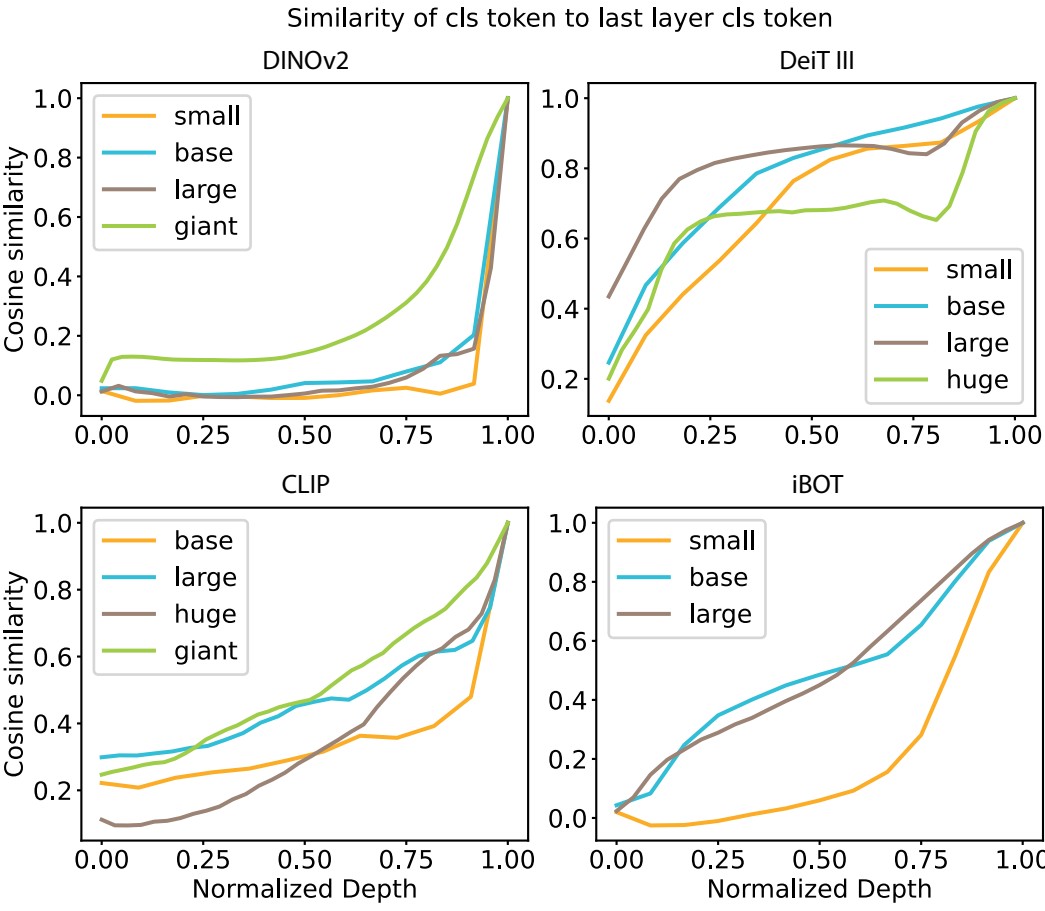

Figure 9: Layerwise similarity between the `[CLS]` token and the last-layer `[CLS]` token as discussed in Section 4. CLIP and iBOT mirror the DINOv2 behavior, whereas DeiT III does not exhibit a clear relationship between model size and early `[CLS]` integration. Note that DeiT III is the only model trained with drop-path regularization, which may explain why all model sizes integrate final output information early in the hierarchy.

