# OpenReview forum: "Register and [CLS] tokens induce a decoupling of local and global features in large ViTs"
_NeurIPS.cc/2025/Conference — NeurIPS 2025 poster_

### Official Review · Reviewer_WKgH · 2025-06-23

**Clarity:** 3
**Significance:** 1
**Originality:** 2
**Rating:** 3
**Confidence:** 3

**Summary:**

The authors analyze the behavior of register tokens across ViT models of different sizes, highlighting that in register-based ViT models, the patch integration assumption does not hold. In addition, they provide an analysis based on skip connections, showing that register or [CLS] tokens can ignore the local information of patch tokens via skip connections.

**Questions:**

1. All analyses in the paper appear to be entirely based on DINOv2. It remains unclear whether similar phenomena occur in other ViT models. In contrast, the original work [17] conducted experiments across a variety of ViTs.
2. In lines 123–125, the authors emphasize that there is large preference variance. Could the authors provide visualizations or a brief analysis to clarify which types of samples, in the largest model, tend to favor register tokens? This would help to better understand under what circumstances the model deviates from the patch integration assumption.
3. Do register tokens exhibit similar behavior in shallow ViT networks？

**Ethical Concerns:**

["NO or VERY MINOR ethics concerns only"]

**Final Justification:**

The paper analyzes large ViT models with a register and provides strong evidence challenging the “patch integration assumption,” along with an interesting analysis of the [CLS] skip connection in an attempt to explain why the patch integration assumption fails. However, since the register inherently carries high-norm global information, I believe it is expected that the aggregation of patch tokens (low-norm) by the CLS token cannot directly correspond to the global information in the representation — and this is the core conclusion of the paper. Therefore, I consider the contribution to remain somewhat incremental and my final score is borderline reject.

**Limitations:**

yes

**Paper Formatting Concerns:**

No formatting issues.

**Quality:**

2

**Strengths And Weaknesses:**

Strengths:
1. The analysis based on skip connections is quite interesting.
2. The authors’ work serves as strong evidence to challenge the “patch integration assumption.”

Weaknesses:
1. The core conclusions of the paper appear to be a straightforward extension of prior work [17], without introducing fundamentally new insights. The paper seems more like an effort to find empirical evidence for some intuitive conclusions:
- Lines 12–13 state that “the CLS token can be interpreted as a register,” but register tokens are essentially an extension of the CLS token as discussed in [17], so this symmetric conclusion seems rather obvious.
- In [17], register tokens are used to store global information to obtain better dense representations and cleaner attention maps, with emphasis on improvements in downstream dense prediction tasks. Given this, it’s already expected that the attention maps from the CLS token to patch tokens would not fully express global information, since part of it is stored in register tokens.
2. The experiments appear to be limited only to the DINOv2 ViT model.
3. The analysis seems confined to only the last few layers of the ViT architecture.

---

> ### Author Rebuttal · Authors · 2025-07-31
>
> Thank you for your insightful review, and the helpful comments on this manuscript.
>
> Please allow us to comment on your suggestions:
>
> > The core conclusions of the paper appear to be a straightforward extension of prior work [17], without introducing fundamentally new insights. The paper seems more like an effort to find empirical evidence for some intuitive conclusions.
>
> Our work does build heavily on [17], but we do believe that our findings are sufficiently distinct from previous work. In particular, we study the effects of the methodology proposed in [17], which are not discussed in that paper. The main findings that set our contribution apart in our opinion are:
>
> - large DINOv2 models rely on register tokens to a surprisingly high degree for forming their [CLS] output; there is a clear relationship between this effect and model capacity. While previous work showed that high-norm tokens seem to encode more global image information, the contribution of high norm tokens to downstream information integration was not studied. The computational role and relevance for [CLS] output of register tokens has not been studied before either.
> - previous work has yielded disagreement on whether high-norm tokens encode global image information ([17], [SINDER]); we show that this is due to methodological differences. Further, we show that high-norm register tokens live primarily in a small subspace which is downscaled during the layer-norm operation, explaining diverging findings and shedding light on the structure of these outliers
> - the attention maps are not as reliable as assumed (see later point)
> - the [CLS] token itself can produce the same effects as the register tokens
> - We also explicitly connect findings for high-norm and register tokens to literature on overparameterization (see also additional results described below)
>
> > - Lines 12–13 state that “the CLS token can be interpreted as a register,” but register tokens are essentially an extension of the CLS token as discussed in [17], so this symmetric conclusion seems rather obvious.
>
> We agree that interpreting the [CLS] token as a register is not a deep theoretical insight. However, we have not seen this interpretation elsewhere and are not sure where to find this being discussed in [17]. Their analysis also does not seem to include the [CLS] token as a register - on the contrary, experiments in [17] (e.g. Fig. 8) include models with 0 register tokens even though a [CLS] token _is_ included. In any case, we would argue that the referenced statement serves rather as a device for unfolding the story of the paper and connecting the findings from section 3 and 4 rather than a theoretical contribution.
>
> > - In [17], register tokens are used to store global information to obtain better dense representations and cleaner attention maps, with emphasis on improvements in downstream dense prediction tasks. Given this, it’s already expected that the attention maps from the CLS token to patch tokens would not fully express global information, since part of it is stored in register tokens.
>
> We do agree, however the degree to which global image information integration is pushed into these register tokens was, in our opinion, not obvious and has not been studied before. Of course one would expect some information contribution from the registers, but that [CLS] output and output computed from attention are in many cases completely orthogonal was striking to us. It has been claimed multiple times that attention maps from register models are mote interpretable, see for example
> 'Register tokens enable interpretable attention maps in all vision transformers...' [17],
> 'Shifting the outlier tokens outside of the image area mimics register behavior at test-time (...), resulting in more interpretable attention patterns' [101],
> 'Integrating registers has been shown to improve interpretability...' [102]
>
> Therefore, we argue it is not clear to the field that attention maps from register models do not reflect global information integration. We believe that this adds strong to merit to our work, as it seems important for the community to be aware of this potential pitfall.
>
>
> > The experiments appear to be limited only to the DINOv2 ViT model.
>
> We agree that this is a shortcoming of our work and have added further experiments to improve our contribution. First, the reason why the study on register models focuses only on DINOv2 is simply that, to the best of our knowledge, no other models with register tokens have been released openly. The paper on DINOv2 with registers [17] has been cited over 500 times and the model is being used by the community which we believe warrants an interest in this model specifically. Further, due to training cost, other work on register tokens also focuses on DINOv2 rather than retraining big models from scratch ([101], [102]).
>
> To connect findings between models with register tokens and those only employing a [CLS] token, we also studied DINOv2 in the second part of the manuscript which does not rely on register tokens. We do agree with your comment that this approach needlessly narrows down the architecture coverage. To remedy this, we additionally ran the experiments in Section 4 for OpenCLIP and DeiTIII to cover contrastive language training and supervised training as well. Unfortunately, NeurIPS doesn't allow plots in the rebuttal this year - however, the findings mirror the ones for DINOv2: Attention to the skip connection grows dramatically with model size in all model families, and similarity between full model output and model output based on patch features declines with model size for all models. Therefore, the findings shown in Fig. 5,6 do seem to be representative for a variety of training schemes. Interestingly, as hypothesized in section 3.5, we observe that the tiny DeiTIII model trained on Imagenet1k shows similar metrics to the giant one trained on Imagenet22k, indicating that high attention to the skip connection arises from model capacity relative to complexity of the training data/task rather than model complexity itself.
>
> > The analysis seems confined to only the last few layers of the ViT architecture.
>
> While section 4 includes a brief explanation for how the [CLS] representation emerges throughout the model, we did focus on the last model layer to investigate how the final global representation is formed. We argue that the last layer is most interesting for our purposes because
> a) the resulting local and global features are the ones usually used for downstream tasks
> b) (the attention map of) this layer was studied in previous works
> To address your comment, we ran an additional experiment on how attention to patch and register tokens is distributed at each model layer. We will include this in the appendix of the manuscript. In brief, we find that models of all sizes attend to register tokens in intermediate layers to varying degrees. A very clear picture arises only at the last layer, where we find the monotonic relationship between model size and attention to register tokens when the final output is formed.
>
> > 1. All analyses in the paper appear to be entirely based on DINOv2. It remains unclear whether similar phenomena occur in other ViT models. In contrast, the original work [17] conducted experiments across a variety of ViTs.
>
> We hope that this is satisfyingly addressed above - if there are further questions, please let us know.
>
> > 1. In lines 123–125, the authors emphasize that there is large preference variance. Could the authors provide visualizations or a brief analysis to clarify which types of samples, in the largest model, tend to favor register tokens? This would help to better understand under what circumstances the model deviates from the patch integration assumption.
>
> This is a very good point, and we also spent some time trying to find structure as to which types of samples were processed primarily in the register tokens. However, this analysis did not lead to any clear conclusions - images of a variety of classes lead to strong violations of the patch integration assumption.
>
> > Do register tokens exhibit similar behavior in shallow ViT networks？
>
> From our experiments, it does seem like these behaviors only arise in the larger models. The 'small' and 'base' model _are_ more shallow than the 'large' and 'giant' model, but of course the depth is confounded with total parameter count. It would be interesting to study these effects in models with different depth or width while keeping the numbers of parameters fixed. However, at this point we can only tie the effects to the number of parameters.
>
> Again, thanks for your review. We hope that our comments and additional work have sufficiently answered some of your concerns and increased your confidence in our work.
>
> [17] Vision Transformers Need Registers
>
> [101] Vision Transformers Don’t Need Trained Registers
>
> [102] Registers in Small Vision Transformers

---

> > ### Comment · Reviewer_WKgH · 2025-08-04
> >
> > Thank you to the authors for the detailed response, which has clarified many of my concerns. I still have a few questions:
> > 1. Regarding the statement ([CLS] can be interpreted as a register), the authors acknowledge that: “In any case, we would argue that the referenced statement serves rather as a device for unfolding the story of the paper and connecting the findings from section 3 and 4 rather than a theoretical contribution.” Given this,  I think such a statement should not appear in the abstract, as it may mislead readers about the actual contributions of the paper.
> > 2. Prior works [17, 101, 102] emphasize that register tokens improve interpretability, but the notion of "improved interpretability" there does not seem to refer to how global information is aggregated. This *interpretability* does not aim to explain *how global information is obtained*. Although the attention scores from the [CLS] token to patch tokens are relatively low compared to register tokens, the relative attention scores among patch tokens can still reflect their importance. Under the premise that  some global information is aggregated by the register tokens, it seems foreseeable that the features aggregated by the [CLS] token over patch tokens cannot directly correspond to the global information in the representation. Therefore, this may not be a pitfall or point that requires strong emphasis or clarification for the community.
> >
> > I appreciate the authors’ clarifications and will raise my score. However, I believe the contribution remains somewhat incremental.

---

> > > ### Author Response · Authors · 2025-08-06
> > >
> > > Thank you for taking the time to review our response to the issues you've raised. We are glad to hear that we were able to address most of your concerns and increase your confidence in this work.
> > >
> > > Also, thank you for the interesting discussion in the two remaining comments:
> > >
> > > 1. We see your point that the statement in the abstract may imply that the relationship between register and [CLS] tokens is a major contribution of the paper. Therefore, we have removed this part of the sentence which now reads 'Inspired by these findings, we show that the [CLS] token itself leads to a very similar phenomenon in models without explicit register tokens.' Thank you for the concrete and actionable suggestion.
> > >
> > > 2. We do agree that these prior works do not explicitly claim that the mechanism of global information integration is more interpretable, as the concept of 'interpretability' is not further defined or explored there. However, we would argue that studying the attention maps of the last-layer [CLS] token implicitly always relates to global information integration, as this is the purpose of the [CLS] token. After all, if one were not interested in global information integration, one could also study the attention maps of any of the other (patch) tokens instead.
> > >
> > > Regarding the point,
> > > > ...it seems foreseeable that the features aggregated by the [CLS] token over patch tokens cannot directly correspond to the global information in the representation,
> > >
> > > we do agree, but would point out that the _degree_ to which global information is extracted from registers is of importance - for example, the attention maps of smaller models with registers _do_ accurately reflect global information integration according to our experiments. Even though we had the same prior assumptions that you mention here, we were surprised to see that in some cases there is _no_ correspondence between patch features global features.
> > >
> > > Finally:
> > > > Although the attention scores from the [CLS] token to patch tokens are relatively low compared to register tokens, the relative attention scores among patch tokens can still reflect their importance.
> > >
> > > This is a valid point, however one should emphasize the 'can', here. In the extreme case of orthogonality between patch-extracted features and global features, this statement may not hold. That being said, we do not try claim that the attention maps of DINOv2 with registers are not an improvement over the model without registers, or that the inclusion of registers has no merit. For the application of LOST, for example, [17] demonstrates a clear improvement over the model without registers.
> > >
> > > We do understand and appreciate your argumentation in 2. but seem to have a slightly different view on the implications of this specific point.
> > >
> > > Thank you again for the discussion and the suggestions, which we believe have strengthened the manuscript.

---

### Official Review · Reviewer_Cr6A · 2025-07-03

**Clarity:** 3
**Significance:** 3
**Originality:** 3
**Rating:** 5
**Confidence:** 3

**Summary:**

The paper studies the observations from Darcet et al. (Vision Transformers Need Registers) further in depth. They introduce the term “patch integration assumption” which describes the theory that the [CLS] token learns to select the final features from the most relevant patches for global image understanding. The authors show that this assumption holds for smaller models but not for larger models (ViT-H, Vit-g) through multiple experiments. They hypothesize that: “ larger ViT variants are overparameterized to the point at which DINOv2 training yields simplistic representations at the last layer, relying on a small number of tokens in which global information has already been integrated.”. Further, they also show that in models without registers the skip connections on the CLS token are used to store global information.

**Questions:**

* I find this hypothesis: “we hypothesize that larger ViT variants are overparameterized to  the point at which DINOv2 training yields simplistic representations at the last layer, relying on a small number of tokens in which global information has already been integrated.” interesting. Did you inspect earlier layers in the larger models if maybe the patch integration assumption holds more there and these global aggregated features are simply passed on?

* Why do we want the patch integration assumption to hold? To get better interpretable models? Why can we not use proper explanation methods?

**Ethical Concerns:**

["NO or VERY MINOR ethics concerns only"]

**Final Justification:**

I definitely would advocate for accepting this paper as already outlined by my initial score as it provides quite interesting insights to the computer vision community!

However, as some parts could benefit from clearer discussion (e.g. do we want the patch integration assumption to hold and why is this case for smaller models from the beginning) and more zooming into the evolution of interaction between global and local tokens throughout the model (beyond Fig. 5b), I don't feel strongly enough to raise my score to a strong accept.

**Limitations:**

yes

**Quality:**

3

**Strengths And Weaknesses:**

**Strengths**

* They introduce the term “patch integration assumption” which poses an interesting question, is described well and picked up again and again throughout the paper.
* The paper is embedded well into related work.
* The disconnection between local and global features is studied from multiple angles (attention distribution, CKA with global representations, accuracy when only aggregating from one set).
* The experiments are well designed and hypotheses and conclusions drawn are easy to follow.
* The results are nicely connected to works on overparameterization (section 3.5).


**Weaknesses**

* The study focuses only on the representations and attention weights at the last layer. Especially as the hypothesis is that for larger models, global information gets aggregated earlier, it would have been great to also analyze the attention weights in earlier layers and how this is progressing throughout the model.
* In the Explainable AI literature, it is well known that attention is not a good explanation [1, 2, 3]. This connection is missing in this paper and it also limits the insights/novelty of the paper a bit.
* It should be better discussed why the patch integration assumption should hold (There is not really something in the model to encourage this).

[1] Wiegreffe, Sarah, and Yuval Pinter. "Attention is not not explanation." arXiv preprint arXiv:1908.04626 (2019).

[2] Jain, Sarthak, and Byron C. Wallace. "Attention is not explanation." arXiv preprint arXiv:1902.10186 (2019).

[3] Ali, Ameen, et al. "XAI for transformers: Better explanations through conservative propagation." International conference on machine learning. PMLR, 2022.

---

> ### Author Rebuttal · Authors · 2025-07-31
>
> Thank you for your insightful review and expressing your support for our work. While we believe that this is a fair evaluation, please allow us to briefly address the points of criticism and questions you've raised:
>
> > - The study focuses only on the representations and attention weights at the last layer. Especially as the hypothesis is that for larger models, global information gets aggregated earlier, it would have been great to also analyze the attention weights in earlier layers and how this is progressing throughout the model.
>
> Our reasoning for focusing on the last layer was that
> a) The attention maps of this layer were studied in previous work
> b) Usually both dense and global features are extracted from this layer for downstream tasks - therefore we believe that the integration of local information is most relevant at this layer.
> Regarding intermediate layers, for models without explicit register tokens we did  analyze how the [CLS] representations evolve with depth in Fig. 5b) which is one step in the direction you've raised.
> Further, in response to your comment, we've run an additional experiment on DINOv2 with registers, measuring the ['CLS] attention to patch and register tokens at each layer. Interestingly, the [CLS] token in models of all sizes attends to both patch and register tokens in intermediate layers. The clear relationship between model size and attention to registers seems to emerge only at the last layer. Subsequent work should study this phenomenon in depth - it should be noted though that the intermediate [CLS] tokens are of course not directly trained to extract global information, which is why their attention patterns may differ from the last layer.
>
> > - In the Explainable AI literature, it is well known that attention is not a good explanation [1, 2, 3]. This connection is missing in this paper and it also limits the insights/novelty of the paper a bit.
>
> We do agree with you statement, however we would argue that our analysis of the attention maps still has merit to the field, as several works have claimed that attention maps of models with registers _are_ more interpretable, see for example:
>
> 'Register tokens enable interpretable attention maps in all vision transformers...' [17],
> 'Shifting the outlier tokens outside of the image area mimics register behavior at test-time (...), resulting in more interpretable attention patterns' [101],
> 'Integrating registers has been shown to improve interpretability...' [102]
>
> Therefore, while the downsides of studying attention maps have been studied as a whole, we believe that the field should also be aware of the specific pitfalls associated with attention maps of register models.
>
> > - It should be better discussed why the patch integration assumption should hold (There is not really something in the model to encourage this).
>
> This does pose an interesting reversal of the hypothesis. Our results show empirically that smaller models _do_ satisfy the patch integration assumption (see also additional experiments mentioned below) and previous work has simply assumed that they do implicitly. When considering the original work on high-norm tokens [17] one can indeed consider the models that exhibit outlier tokens to be the 'standard case' and explicitly study those model that do not display them. To the best of our knowledge, analyses of that kind have not been done and we think that this would be a very interesting avenue. For this work, we hope you agree that at least a systematic treatment of this question is out of scope.
>
> > - I find this hypothesis: “we hypothesize that larger ViT variants are overparameterized to the point at which DINOv2 training yields simplistic representations at the last layer, relying on a small number of tokens in which global information has already been integrated.” interesting. Did you inspect earlier layers in the larger models if maybe the patch integration assumption holds more there and these global aggregated features are simply passed on?
>
> For the models without register tokens, the analysis shown in Fig. 5b) does support the claim that in larger models, global feature integration starts earlier in the hierarchy and those features are passed on with small modifications. Therefore, the last layer essentially receives its output as input in the giant model.
> For the model with registers, we did not test this explicitly but we would argue that the results in Fig. 2 also strongly indicate this conclusion. Of course, in the intermediate model layers, the model has to extract all necessary information from the image patches but in the last layer the model attends primarily to just 5 register tokens (compared to 256 patch tokens!). A layer-wise analysis like the one shown in Fig. 5b) probing whether the information stored in register tokens saturates with depth would also be interesting here, but unfortunately due to time constraints we were not able to implement this during the rebuttal phase.
>
> >- Why do we want the patch integration assumption to hold? To get better interpretable models? Why can we not use proper explanation methods?
>
> First, we would argue that the simplicity of being able to write the [CLS] output as a convex combination of **meaningful** patch features is itself an advantage over more involved attribution methods because, it is very intuitive and easy to apply/understand for the user.
> Beyond interpretability, joint training of dense and global features as in DINOv2 allows the user to explicitly ground the global representation in the dense feature maps as done in [15, 16]. More generally, we are referring here to tasks that rely on a correspondence between dense local features and the global image embedding.
>
> ---
>
> Finally, please note that we've also run additional experiments including more model architectures for the skip connections studied in Section 4. Since there is no global response this year, and to avoid walls of text, we would kindly refer you to the response to reviewer qxac in case you are interested.
>
> [15] Extract Free Dense Labels from CLIP
>
> [16] BRAIN MAPPING WITH DENSE FEATURES
>
> [17] Vision Transformers Need Registers
>
> [101] Vision Transformers Don’t Need Trained Registers
>
> [102] Registers in Small Vision Transformers

---

> > ### Comment · Reviewer_Cr6A · 2025-08-08
> >
> > Thank you for responding to my comments and running additional experiments!
> >
> >
> >
> > > Therefore, while the downsides of studying attention maps have been studied as a whole, we believe that the field should also be aware of the specific pitfalls associated with attention maps of register models
> >
> > Yes, I agree. I think this connection to the Explainable AI literature should just be pointed out somewhere in the paper!
> >
> > I definitely would advocate for accepting this paper as already outlined by my initial score as it provides quite interesting insights to the computer vision community!
> >
> > However, as some parts could benefit from clearer discussion (e.g. do we want the patch integration assumption to hold and why is this case for smaller models from the beginning) and more zooming into the evolution of interaction between global and local tokens throughout the model (beyond Fig. 5b), I don't feel strongly enough to raise my score to a strong accept.

---

### Official Review · Reviewer_inCN · 2025-07-03

**Clarity:** 3
**Significance:** 3
**Originality:** 4
**Rating:** 5
**Confidence:** 2

**Summary:**

This paper identifies a fundamental issue in ViTs, such as DINOv2: a decoupling between global image representations and local patch features. Traditionally, global features are assumed to be formed by aggregating local patch embeddings via [CLS] token attention. However, the authors show that this assumption breaks down in large models. The authors point out that [CLS] often attends to special register tokens rather than image patches, making the global representation insensitive to local details. Even without explicit register tokens, residual connections allow [CLS] to bypass patch information, effectively acting as an implicit register. Besides, the root cause is attributed to overparameterization, allowing global information to be compressed early and isolated from the image structure.

**Questions:**

The authors may also provide some robustness/variance evaluation on the experiment results.

**Ethical Concerns:**

["NO or VERY MINOR ethics concerns only"]

**Final Justification:**

The authors addressed my concerns, I recommend accept.

**Limitations:**

Yes

**Quality:**

4

**Strengths And Weaknesses:**

The authors present a rigorous and well-structured analysis of the interaction between the [CLS] token and register tokens. The paper is clearly written and easy to follow. I like the authors to identify a fundamental issue—overparameterization—and links it to the phenomenon of neural collapse, offering a novel and insightful perspective. I believe this work can inspire further research toward improving the interpretability and structure of Vision Transformers.

While the authors suggest avoiding residual connections between register tokens and the [CLS] token, it would strengthen the paper to include empirical evidence supporting this claim. Are there any experiments or architectural designs specifically aimed at mitigating this issue? A proof-of-concept evaluation would greatly enhance the contribution. Also as mentioned in the limitation part, it would be important for the community to understand how these discussed relationships can improve the performance of these ViTs.

---

> ### Author Rebuttal · Authors · 2025-07-31
>
> Thank you for you thorough review - we appreciate your confidence in the value of our work. Please allow us to respond to your comment:
>
> > The authors may also provide some robustness/variance evaluation on the experiment results.
>
> To strengthen the empirical results of the paper, we have included two more models for the experiments in Section 4 (DeiTII for supervised training, OpenCLIP for contrastive language training). While, unfortunately, we can not share results visually this year, the findings mirror the ones for DINOv2: Attention to the skip connection grows dramatically with model size in all model families, and similarity between full model output and model output based on patch features declines with model size for all models. Therefore, the findings shown in Fig. 5,6 do seem to be representative for a variety of training schemes, therefore demonstrating the robustness of the findings. Interestingly, as hypothesized in section 3.5, we observe that the tiny DeiTIII model trained on Imagenet1k shows similar metrics to the giant one trained on Imagenet22k, indicating that high attention to the skip connection arises from model capacity relative to complexity of the training data/task rather than model complexity itself.

---

> > ### Comment · Reviewer_inCN · 2025-08-09
> > **Addressed my concerns**
> >
> > Thanks for the rebuttal. The authors addressed my concerns. I maintain the score.

---

### Official Review · Reviewer_qxac · 2025-07-05

**Clarity:** 3
**Significance:** 2
**Originality:** 2
**Rating:** 3
**Confidence:** 4

**Summary:**

This paper investigates the patch integration assumption in Vision Transformers (ViTs), focusing on how register tokens and [CLS] skip connections in large models (e.g., DINOv2) decouple global and local feature representations. Through empirical analysis (e.g., centered kernel alignment, attention distribution probing), the authors demonstrate that register tokens dominate global representations in large ViTs, rendering attention maps unreliable for attribution. The work further links this phenomenon to overparameterization and suggests architectural refinements for interpretability.

**Questions:**

* **(Q1)** Could the authors' findings hold for *non-DINOv2 architectures* (e.g., DeiT, Swin Transformers) or other pretraining methods (e.g., masked image modeling or supervised pre-training)? If not, how might this limit broader applicability?

* **(Q2)** The authors claim register tokens harm dense-task performance (Sec. 3.5), but provide no metrics. Can the authors benchmark patch-feature quality on tasks like segmentation using standard datasets (e.g., ADE20K)?

* **(Q3)** Instead of removing registers/skips (Sec. 5), could pre-norm / post-norm regularization or drop-path regularization (Sec. 4) preserve interpretability?

**Ethical Concerns:**

["NO or VERY MINOR ethics concerns only"]

**Final Justification:**

Overall, this submission offers a careful and useful empirical clarification of when attention maps fail as attributions in large ViTs, including a non-trivial analysis of the [CLS] residual. However, in its current form the evidence base is narrow and the downstream impact is not quantified. If the authors incorporate the rebuttal-reported cross-model experiments and add a small, focused downstream evaluation or residual-gating ablation, I would be inclined to raise my score. I lean Borderline Reject on significance/external validity grounds.

**Limitations:**

The paper acknowledges scope limitations (Sec. 5) and computational/resource constraints (Sec. A). However, methodology limitations (e.g., narrow architecture coverage) require expanded discussion. Suggestion: Explicitly state that conclusions are DINOv2-specific and may not generalize.

**Quality:**

2

**Strengths And Weaknesses:**

### **Strengths**

This work tries to address a critical issue in ViT interpretability and provide recent findings about attention-map artifacts based on DINOv2 and [1]. The authors present several interesting findings based on empirical studies and visualizations. Overall, this manuscript is well-structured with clear visualizations (e.g., Fig. 4 illustrating orthogonal feature spaces).

### **Weaknesses**

This paper does not provide any new designs but conducts empirical studies with several variants of ViTs using different types of special tokens (e.g., CLS tokens or register tokens).

* **(W1)** This work is not contributive enough to be published as a conference paper. Compared to previous work like DINOv2 with register tokens, this work only provides incremental findings (e.g., skip-connection effects) without transformative solutions. It limits the novelty of this work. Such empirical studies should provide some benchmarks or technique contributions. I recommend that the authors resubmit the manuscript to some workshops or the track of findings, or enhance it with more benchmarks or ablation models.

* **(W2)** Drawbacks in the approaches of the empirical studies. On the one hand, this work relies heavily on the analysis methods used in [1] , e.g., feature similarity metrics (CKA), without downstream task evaluation (e.g., segmentation / detection performance). On the other hand, the authors only evaluated the classical Vision Transformers with contrastive learning (e.g., DINO and DINOv2), how about other similar Transformer architectures and different pre-training methods? Based on the limitations and drawbacks above, the provided conclusion and findings might be limited to certain research domains rather than being useful to general vision network designs and pre-training.

### Reference

[1] Vision Transformers Need Registers. ICLR, 2024.

---

> ### Author Rebuttal · Authors · 2025-07-31
>
> Thank you for your comprehensive review and constructive comments on our work.
>
> Please allow us to comment on your suggestions:
>
> >  Compared to previous work like DINOv2 [17] with register tokens, this work only provides incremental findings (e.g., skip-connection effects)...
>
> We would argue that the findings presented in this work are not merely incremental with respect to previous work; we will summarize our opinion on how the main contributions set our work apart from previous efforts. To the best of our knowledge we are the first to show that:
>
> - large DINOv2 models rely on register tokens to a surprisingly high degree for forming their [CLS] output; there is a clear relationship between this effect and model capacity. While previous work showed that high-norm tokens seem to encode more global image information, the contribution of high-norm tokens to downstream information integration was not studied. The computational role and relevance for [CLS] output of register tokens has not been studied before either.
> - previous work has yielded disagreement on whether high-norm tokens encode global image information ([17], [SINDER]); we show that this is due to methodological differences. Further, we show that high-norm register tokens live primarily in a small subspace which is downscaled during the layer-norm operation, explaining diverging findings and shedding light on the structure of these outliers
> - representations based on register/patch tokens are surprisingly orthogonal which, we argue, has important implications for the field. Several recent works [...] have claimed that incorporating register tokens yields more interpretable attention maps:
> 	-'Register tokens enable interpretable attention maps in all vision transformers...' [17],
> 	'Shifting the outlier tokens outside of the image area mimics register behavior at test-time (...), resulting in more interpretable attention patterns' [101],
> 	'Integrating registers has been shown to improve interpretability...' [102]
> 	However, these claims do not hold when the attention map does not reflect the model output in a veridical way. We believe that an awareness of these limitations makes an important contribution to the field.
> - The [CLS] token itself can produce the same effects
> - We also explicitly connect findings for high-norm and register tokens to literature on overparameterization (Please note also additional novel results described below)f
>
> > This work relies heavily on the analysis methods used in [17] , e.g., feature similarity metrics (CKA), without downstream task evaluation (e.g., segmentation / detection performance).
>
> We would kindly point out that, as far as we can tell, CKA and similarity of patch/register representations to the full model output have not been studied in previous work, including [17]. To the best of our knowledge, we are the first to analyze the representations of patch/register tokens using feature similarity metrics. Downstream task performance (e.g. segmenation, LOST) has been studied relatively extensively, and we do not claim that models with registers have poor performance on these tasks. The focus of our contribution is rather on understanding the computational role filled by register tokens, and pointing out potential pitfalls like seemingly more interpretable attention maps.
>
> >  On the other hand, the authors only evaluated the classical Vision Transformers with contrastive learning (e.g., DINO and DINOv2), how about other similar Transformer architectures and different pre-training methods?
>
> This is a very good point and we have tried to address it by running further experiments. First, the reason why the study on register models focuses only on DINOv2 is simply that, to the best of our knowledge, no other large-scale models with register tokens have been released openly. The paper on DINOv2 with registers [17] has been cited over 500 times and the model is being used by the community which we believe warrants an interest in this model specifically. Further, due to training cost, other work on register tokens also focuses on DINOv2 rather than retraining other models from scratch ([102], [102]).
>
> To connect findings between models with register tokens and those only employing a [CLS] token, we also studied DINOv2 in the second part of the manuscript which does not rely on register tokens. We do agree with your comment that this approach needlessly narrows down the architecture coverage. To remedy this, we additionally ran the experiments in Section 4 for OpenCLIP and DeiTIII to cover contrastive language training and supervised training as well. Unfortunately, NeurIPS doesn't allow plots in the rebuttal this year - however, the findings mirror the ones for DINOv2: Attention to the skip connection grows dramatically with model size in all model families, and similarity between full model output and model output based on patch features declines with model size for all models. Therefore, the findings shown in Fig. 5,6 do seem to be representative for a variety of training schemes. Interestingly, as hypothesized in section 3.5, we observe that the tiny DeiTIII model trained on Imagenet1k shows similar metrics to the giant one trained on Imagenet22k, indicating that high attention to the skip connection arises from model capacity relative to complexity of the training data/task rather than model complexity itself.
>
> > - (Q1) Could the authors' findings hold for _non-DINOv2 architectures_ (e.g., DeiT, Swin Transformers) or other pretraining methods (e.g., masked image modeling or supervised pre-training)? If not, how might this limit broader applicability?
>
> We hope to have answered this question above - if part of the question remains, we are of course open to discussing further.
>
> > - (Q2) The authors claim register tokens harm dense-task performance (Sec. 3.5), but provide no metrics. Can the authors benchmark patch-feature quality on tasks like segmentation using standard datasets (e.g., ADE20K)?
>
> We have tried not to claim poor performance of register models on tense tasks as it's been shown that performance on segmentation and localization is quite good. We were rather referring to tasks that require a tight connection between local and global features (e.g. [15], [16]). However, these statements serve more as motivation to our work rather than aiming to provide quantitative evidence. Therefore we do agree that strong claims in that regard are not justified and have made sure there are no overclaims in the manuscript.
>
> > - (Q3) Instead of removing registers/skips (Sec. 5), could pre-norm / post-norm regularization or drop-path regularization (Sec. 4) preserve interpretability?
>
> This is an interesting question and indeed it should be explored how strongly drop-path regularization can inhibit extreme attention to the skip connection at the last layer. The only statement that we can make with certainty is that DeiTIII, which does use linearly increasing drop-path regularization, shows the same patterns as displayed for DINOv2 in the manuscript (see discussion above). Therefore, at least it does not seem to be a 'quick fix' that works by just including it. With your permission, we would include a brief paragraph in the discussion of the manuscript should it be accepted for publication.
>
> Thanks again for your review - we hope that we were able to address some of your concerns and have perhaps increased your confidence in our work. We are of course open to further discussion on the manuscript.
>
> [17] Vision Transformers Need Registers
>
> [101] Vision Transformers Don’t Need Trained Registers
>
> [102] Registers in Small Vision Transformers

---

> > ### Comment · Reviewer_qxac · 2025-08-07
> > **Feedback to Authors' Rebuttal**
> >
> > Thanks for the detailed response and the efforts the authors provided. Some concerns were tackled, while several weaknesses are still not well addressed.
> >
> > * (W1) Further concerns on external validity. Most results are DINOv2-only in the submission, and the authors' rebuttal mentions OpenCLIP/DeiT-III, but these are not in the paper currently. Will the authors include the cross-model figures/tables (e.g., per-model CKA curves and skip-vs-patch norms) plus minimal training details so readers can verify generality? If masked-image modeling (e.g., MAE/IBOT) is out of scope, can the authors state that explicitly to bound claims?
> >
> > * (W2) & (Q2) Downstream tasks evaluation and interpretability are also important. Some possible analysis could be benefit to the overall contribution. For example, can the authors add a small, focused probe (e.g., ADE20K or some localization metrics) comparing patch-only vs. full [CLS] to link your diagnostics to practical consequences?
> >
> > * Additional suggestions: If you integrate the cross-model evidence you ran and add a minimal downstream/ablation check with uncertainty reporting. Add a short per-image analysis explaining when registers dominate.
> >
> > Given that the new evidence is not yet in the rebuttal, I maintain my original score. I am encouraging the authors to further clarify and provide additional quantitative evidence for my further concerns, and I will consider changing my score if these concerns are tackled.

---

> > > ### Author Response · Authors · 2025-08-08
> > >
> > > Thank you for taking the time to review our response. Since the rebuttal deadline is already approaching, we briefly respond while we are implementing the last experiment you asked for.
> > >
> > > 1. We include all further results prompted by the review phase in the final draft of the manuscript, including CKA curves and skip-vs-patch norms for all additional models to demonstrate that the presented results generalize. We did not test the vanilla MAE training procedure as it does not explicitly model global image information; however, we further include results for iBOT trained on Imagenet. Our results here show a slight trend of increasing attention to skip features with increasing size. However, all models attend primarily to patch tokens, and CKA with patch features is low for all modes, indicating that all of them violate the patch integration assumption.
> > >
> > > 2. We are currently still implementing a quantitative experiment for localization. Qualitatively, we now already include the patch-wise cosine similarity with the [CLS] token for a handful of images in the manuscript. These plots show that for the small model with registers, patches overlapping with foreground objects have high cosine similarity with the [CLS] token. This is not the case for the large model, indicating that global information cannot be grounded in local patches as accurately.
> > >
> > > 3. This is an interesting idea which we have experimented with as well - however, we were not able to find clear answers to the question of which kinds of images are processed mainly in the register tokens. However, we will include a small number of example images which very heavily violate the patch integration assumption in the supplement.

---

> > > > ### Author Response · Authors · 2025-08-08
> > > >
> > > > Please excuse the double comment - we would just like to confirm before the discussion deadline approaches, that we have run further evaluation on COCO, probing the localization abilities of the models with register tokens. Specifically, we test whether the global [CLS] information can be grounded in patch features in terms of segmentation ability. We simply compute for each patch the cosine similarity with the [CLS] token, and then use Otsu's method [103] to convert activation maps into binary segmentation masks. We then compute mean IoU with the ground truth masks. We obtain a mean score of 36.5% in the small model, 34.02% in the base model, and 28.26% in the large model. This serves as further quantitative evidence for a disconnect between local and global features in larger models with register tokens, also in terms of downstream task performance.
> > > >
> > > > We'd like to thank you again for your constructive feedback, which has substantially strengthened the manuscript in our opinion.
> > > >
> > > > [103] A threshold selection method from gray-level histograms

---

### Decision · Program_Chairs · 2025-09-17

**Decision:**

Accept (poster)

**Comment:**

The paper analyzes the representations learned by vision transformers (ViTs) of different sizes with registers or CLS tokens, in particular with respect to the "patch integration assumption" that the final representation is aggregated from patch representations, and attention maps are somewhat representative of the importance of different image regions.

Based on the reviews, author/reviewer discussion, and the paper itself, here are the key strengths and weaknesses.

Strengths
1. Interesting and well-executed analysis, from several different angles
2. Good presentation

Weaknesses
1. The analysis is limited to one model type (DINOv2). Authors mentioned other model types in the rebuttal, but it's not in the paper atm
2. The analysis is limited to the last layer
3. No clear practical conclusions, no application experiments demonstrating useful takeaways from the analysis

Overall, the paper is fairly borderline, with some interesting analysis, but also clearly limited in some ways. This is also reflected by review scores.

Given how the paper is well written and the results are thought-provoking and potentially of interest to the broad community of computer vision researchers, I lean towards recommending acceptance. However, I do very much encourage the authors to improve the paper.